# The Assessment of Postural–Motor, Coordination, and Reflex Functions in Children and Adolescents with a History of Premature Verticalization and Ontogeny Disorders in Their First Year of Life

**DOI:** 10.3390/children11091071

**Published:** 2024-08-31

**Authors:** Mieczysław Maciak, Kamil Koszela, Anna Beniuk, Marta Woldańska-Okońska

**Affiliations:** 1Center for Therapeutic Rehabilitation in Świdnica, 3 Rotmistrza Witolda Pileckiego Street, 58-100 Świdnica, Poland; maciak.wroclaw@gmail.com (M.M.);; 2Neuroorthopedics and Neurology Clinic and Polyclinic, National Institute of Geriatrics, Rheumatology and Rehabilitation, 02-637 Warsaw, Poland; 3Department of Internal Medicine, Rehabilitation and Physical Medicine, Medical University of Lodz, 90-419 Lodz, Poland

**Keywords:** developmental milestones, faulty posture, childcare, primitive reflexes

## Abstract

(1) Background: Contracting diseases or being exposed to adverse environmental factors in the first year of life may impair the development of body posture and motor coordination. The purpose of this study was to evaluate the correlation between data on the speed of passive verticalization, the number of risk factors and the quality of development in the first year of life, and the results of the functional examination of these individuals in adolescence. (2) Methods: Two groups of 60 volunteers, aged 9–14 years, were examined by performing functional tests and the retrospective analysis of their development up to the age of 1 year. The first group consisted of patients diagnosed with postural defects. The control group consisted of healthy people of the same age who volunteered for this study. (3) Results: Statistical analysis showed statistically significant differences between groups in terms of posture (*p* = 0.001), motor coordination (*p* = 0.001), and accumulated primitive reflexes (*p* = 0.001), as well as a high correlation between these disorders and the quality of development in the first year of life. In the first group, for the ages of 3–6 months (r = 0.96; *p* = 0.001), 6–9 months (r = 0.871; *p* = 0.001), and 9–12 months (r = 0.806; *p* = 0.001), no significant correlations were found with the age of 0–3 months. In the second group, the results were as follows: 0–3 months (r = 0.748; *p* = 0.001), 3–6 months (r = 0.862 *p* = 0.001), 6–9 months (r = 0.698; *p* = 0.001), and 9–12 months (r = 0.740; *p* = 0.001). In the group of adolescents with posture defects, we observed an earlier time of passive verticalization and sitting, as well as more frequent use of loungers, seats, and walkers (*p* = 0.026). (4) Conclusions: The analysis of this study’s data indicates that the development of body posture and motor coordination may be impaired due to accelerated and passive verticalization in the first year of life.

## 1. Introduction

The subject of this work is the mechanisms influencing the appearance of milestones in the first year of life and their direct impact on the quality of body posture and motor coordination in later periods of life. Psychomotor functions (developmental milestones) are acquired step by step in the first year of life, are permanent, and can be assessed in later periods of life [1,2]. Active antigravity locomotion, in which the vestibular organ plays a leading role, may be an important mechanism, integrating immature postnatal neural networks of the spinal cord, subcortex, and cerebral cortex [2]. The reason for designing this study was the observation of the different developmental trajectories of children who did not suffer from chronic diseases and in whom the influence of perinatal factors was not proven. It was hypothesized that the probable cause of their poor development and postural disorders was unfavorable environmental conditions in the first year of life. Among the risk factors, attention was drawn to the incorrect habits of passive and premature verticalization, which may be common and potentially correctable causes. Due to the developmental immaturity of the balance system in newborns and infants, passive verticalization can stimulate and perpetuate primitive reflexes and contribute to the disruption of the developmental path in terms of body posture and motor coordination. Care errors from this period may have lasting consequences for the child’s further development. Therefore, the assessment of motor development is an important element of care for children in the first year of life, as well as for people of all ages [3,4,5]. Motor development shapes the integrity of the neuromuscular system and determines the quality of other higher-order functions. Motor development disorders may be associated with coexisting problems with cognitive, emotional, and speech functions [6,7,8,9]. Various scales are used to assess the course of developmental milestones [10,11,12,13,14]. 

By providing the child with a safe space and freedom of movement, we enable the optimal development of motor functions [15,16]. The balance organ, in which vestibular stimuli automatically become multisensory and multimodal, plays a leading role in the integration of senses during development [17,18,19]. The central nervous system processes vestibular, proprioceptive, and visual stimuli, shaping the integrated response of the neuromuscular system to stabilize the body in the gravity field. The proper stimulation of the vestibular organ affects the development of vision and body posture. Therefore, stable activating positions in subsequent stages of verticalization stimulate the achievement of further developmental milestones, which should be carefully monitored and corrected as part of developmental supervision [20,21,22,23]. In the first six months of life, the child responds to environmental stimuli owing to newborn primitive reflexes [24,25,26]. Normal development in the first year of life is manifested in the inhibition of primitive reflexes and the formation of coordinated psychomotor functions [27,28]. Diseases or unfavorable developmental conditions may result in the incomplete integration of reflexes [29,30,31]. There is evidence that the accumulation of abnormal reflexes predicts the failure to achieve developmental milestones within a typical timeframe and the occurrence of disorders of central psychomotor coordination [32,33,34]. The diagnosis of a child’s development should combine the assessment of the inhibition of primitive reflexes, the occurrence of subsequent developmental milestones, and the co-occurrence of risk factors. The detected abnormalities require further specialist diagnostic treatment [35,36,37,38]. There are inconsistencies in data regarding the standards for achieving developmental milestones and the principles of infant care in relation to the stage of development [39]. 

For the purposes of this study, the development of a child in the first year of life was divided into four stages, which were characterized by their duration (approximately 3 months), their staged nature, and the achievement of motor skills at the end of each stage (so-called developmental positions). The assessment of these developmental positions in subsequent stages may indicate their quality of development. In the first stage, after 3 months of life, we expect a symmetrical and axial positioning of the head, torso, and pelvis. In the supine position, the hip and shoulder joints should be centrally positioned, with the limbs held in front of the body. In the prone position, the child supports itself on its elbows and keeps its head along the axis of the torso. In the second stage, seen at around 6 months, the child acquires the ability to coordinate rotation, with body stabilization obtained by 3 months. In the third stage, spanning up to 9 months of age, the child acquires the ability to actively crawl and three-point oblique sitting thanks to the symmetrical stabilization of the body from the first stage and the rotation function of the second stage of development. In the fourth stage (9–12 months), the child sits independently with a straight back, manipulating its hands free of support. Then, the child stands up by kneeling with support and begins walking using furniture. The development of independent walking, squatting, and balance will continue until the 18th month [10]. Development positions can be studied during their development in infants and later in life. Abnormalities in their performances give us a picture of the course of developmental milestones in an infant’s first year of life.

The aim of this study was to assess the influence of premature passive verticalization and perinatal risk factors on the course of developmental milestones in the first year of life and on the development of body posture, motor coordination, and primitive reflexes at the age of 9–14 years. The assessment of body balance, consisting of performing, for example, the one-legged standing test, the alternating leg test, and the Romberg test using posturography, showed the degree of maturity of the nervous systems of the studied persons [18,19]. This study used visual motor development charts to conduct functional tests and facilitate the retrospective analysis of developmental stages in the first year of life.

## 2. Materials and Methods

This study was designed to examine 120 individuals aged 9–14 years, with particular emphasis on functional tests and assessments using a posturograph and visual motor development charts. In addition, medical records were reviewed, and a medical interview was conducted to assess motor development and the time of passive verticalization in the first year of life. Retrospective data on developmental milestones and care techniques during the first year of life were obtained from medical records and interviews with caregivers during this period. Visual charts of child development during the first year of life were used to assist caregivers in answering questions. Photographs taken during this time were also analyzed to assess the achievement of motor milestones. The results of the tests and interview were recorded in a questionnaire according to the format that was submitted as Appendix 1 to the application for approval by the Bioethics Committee (Table 1).

The results of tests and procedures were entered into Table 1 using the following scoring system:Correctness.Slight disorder/reflex.Distinct disorder/reflex.

In the case of a score of 2 or 3, an additional description of abnormalities was provided with the active participation of the study participants and their caretakers. The scoring, estimating the quality of development in the individual stages of the first year of life, was based on the assessment of information obtained from parents after becoming acquainted with the patients’ medical records. All tests were performed by the same diagnostician using the same methodology (a total of 28 tested parameters). 

### 2.1. Participants

A group of 120 participants aged 9–14 were divided into two groups of 60 individuals. Participants in group 1 of this study were recruited from among patients referred to a posture treatment clinic due to body shape and movement coordination disorders, including scoliosis, deep kyphosis, valgus, or varus limb deformity. The second cohort, meaning the comparison group, consisted of individuals in the same age range who were free from chronic diseases and who had voluntarily attended a postural assessment as part of a preventive examination. These participants were recruited for this study through announcements made in local primary schools. The inclusion criteria for the first group were the presence of a diagnosed postural defect and an age range of 9 to 14 years. The exclusion criteria included severe congenital defects, genetic, neurological, and metabolic disorders that would prevent the performance of functional tests, as well as other conditions that could impact psychomotor development. For the comparison group, the inclusion criteria were an age of 9–14 years and attendance at a mainstream school, while the exclusion criteria were the same as for the first group: severe congenital defects; genetic, neurological, and metabolic disorders; and other issues that could prevent the performance of functional tests.

Study participants:-group 1 (*n* = 60);-group 2 (*n* = 60).

The method of selection was convenience sampling.

The characteristics of the study groups were assessed (Table 2) and both study groups (group 1 and group 2) were comparable in terms of gender and age.

The study’s design was also based on the hypothesis that the successive stages of a child’s development in the first year of life shape the basic structures and functions of the body for life, which will then only be expanded upon and improved. For this reason, young people from a wide age range of 9–14 years qualified for the study, allowing us to include periods of accelerated and decelerated body growth.

### 2.2. Research Tools

In the study (Table 1), items 6 and 7 of the questionnaire assessed the pattern of persistent asymmetric tonic neck reflex (ATNR) and symmetric tonic neck reflex (STNR). These are primitive neuromuscular reflexes present in the early development of a child, and should be integrated into the process of the maturation of the nervous system in the first year of life. These reflexes should no longer be observed in subjects aged 9–14 years. The delayed integration of these reflexes may affect the course of milestones and disturb the child’s psychomotor development [32,33,34,36].

In order to improve the validity and reliability of the tests performed and the developmental data collected, we used visual charts of developmental positions in the first year of life, designed according to the principles of developmental kinesiology—the so-called dynamic neuromuscular stabilization (DNS) outlined in the work of Kolar [2]. Developmental positions, from 3 months to 13 months, are demonstrated in both infants and adults. The charts demonstrate the basic principles of postural stabilization, ipsilateral and contralateral locomotion patterns, and open and closed kinematic chains. These positions can be assessed and rehabilitated during their development in infancy and when they are established in later periods of life. The correct performance of these tests may indicate the maturity of the nervous and musculoskeletal systems. The DNS chart model was attached to the application to gain the consent of the Bioethics Committee as Annex No. 7. Below, we present selected developmental positions from the first year of life, the quality of which can also be assessed in later periods of life (Figure 1).

### 2.3. Procedures

This study was approved by the Bioethics Committee at the Medical University in Lodz. (No: RNN/422/18/KE, 10 December 2018). The research project consisted of performing functional tests and posturography tests in adolescents, aged 9–14 years, to assess their body posture and motor skills. Furthermore, a questionnaire was completed regarding development in the first year of life. This was based on the interview and medical records. Participation in this study was voluntary and free of charge, with the possibility of refusing or withdrawing consent at any time without giving a reason and without any consequences, thus maintaining the right to treatment in the same center. The research study was not associated with any health risks. Functional tests were performed after obtaining informed consent and in the presence of the study participant’s legal guardian. The documentation is stored in a confidential manner, consistent with the principles of personal data processing.

No personal data were used in the description of this study. This study was performed over a 12-month period at the Center for Therapeutic Rehabilitation (Świdnica 58-100, 3 Rotmistrza Witolda Pileckiego St., Poland), which has civil liability insurance. Each participant and their legal guardian received detailed information about this study and performed tests. The study’s participants received a copy of information about this study and could ask questions, to which sufficient answers were provided. There was a benefit for subjects and their guardians in the form of a detailed examination and, if any abnormalities were found, explanations of recommendations for further rehabilitation proceedings were given. There were no adverse effects of involvement in the study.

Each patient was assessed for current and past illnesses, medications, physical activity, and educational problems. Body posture in a standing position was assessed for deviations from anatomical norms. Then, the patient was instructed to adopt developmental positions appropriate to those achieved in the subsequent stages of development in the first year of life. Visual tables of motor development were used for this purpose. The alternating test assessed the alternation of the upper and lower limbs while walking in place. In order to objectify the body posture, balance, and distribution of foot pressure on the ground, a device was used—the Fee Step platform—which assessed the reactions of ground forces in static and dynamic conditions. A digital record of the test result is contained in the patient’s documentation.

### 2.4. Subjective Assessment

This study’s design relied on the subjective evaluation of the tests. However, in order to improve the validity and reliability of the tests, the study was based on performing detailed functional tests, conducted by the examining physician in cooperation with the participants and their caregivers. The scores of the performed tests, ranging from 1 to 3, were given jointly after discussion. Visual charts of motor development were used to objectify the quality of the developmental positions and postural tests performed and to note the time taken to achieve developmental milestones (Figure 1). In addition, the retrospective assessment of the child’s developmental milestones used data from submitted medical records, archival photographs, and information collected from the child’s caregivers. To objectify the results of the balance assessment in the Romberg test and the quality of foot loading, the Free-Step digital posturograph was used.

### 2.5. Statistical Analysis

MS Excel 2019 was used for data collection. Statistical analysis was performed using the Statistica 13 package. Data were normalized by standardization to ensure the comparability of the results. Student’s *t* tests were used to compare means between groups, and the effect size was assessed using Cohen’s d coefficient. The results indicate significant differences between groups (*p* < 0.05). The average effect size for all performed tests, about 1.00, suggests that the effects have practical importance. 

## 3. Results

The figure below (Figure 2) shows the distribution of body posture assessment results for a standing position in both groups.

Statistical analysis with Statistica-13 (statistical significance threshold *p* ≤ 0.05; both groups, Group 1: N = 60, Group 2: N = 60, were comparable in terms of gender and age) showed statistically significant correlations between data obtained from the examination of body functions and features in adolescence and the course of milestones in these subjects during the first year of life (Table 3).

High correlations were found in the study group (Table 4): for the ages of 3–6 months (r = 0.96 *p* < 0.001), 6–9 months (r = 0.871 *p* < 0.001), and 9–12 months (r = 0.806 *p* < 0.001), no significant correlations were found with the age of 0–3 months. The control group also showed a high correlation in the investigated age ranges: 0–3 months (r = 0.748 *p* < 0.001), 3–6 months (r = 0.862 *p* < 0.001), 6–9 months (r = 0.698 *p* < 0.001), and 9–12 months (r = 0.740 *p* < 0.001).

Statistically significant differences were observed between Groups 1 and 2 (*p* ≤ 0.05) in terms of the values and distributions of the above-mentioned parameters and in terms of the following:Body posture, motor coordination and the presence of persistent primitive reflexes;The way of carrying and caring and functions achieved at developmental milestones.

No statistically significant differences were observed between the groups in terms of the following:The presence of risks and diseases in the perinatal period;The number of Apgar scores;Rehabilitation treatment (comprehensive);Used rehabilitation/orthopedic aids (e.g., abduction braces);Comorbidities of developmental age.

The statistical significance of the differences between groups is shown in Table 5.

This study also assessed the age of assuming a sitting position and walking in both investigated groups (Table 6).

In the group of adolescents with faulty posture, earlier sitting and standing times and the more frequent use of loungers, seats, and walkers were observed.

## 4. Discussion

The aim of this study was to assess which environmental conditions in a child’s first year of life may have the greatest impact on the development of posture and coordination. The original assumption of the work was based on the data from the literature showing that a child’s development in the first year of life proceeds in an orderly manner in successive stages. Each stage should appear in a specific order and last long enough for the developmental milestones to appear. They can be assessed in terms of movement as so-called developmental positions, considering quality, order, and the time of appearance [1,2]. Testing identical developmental positions in adolescence allows for the assessment of acquired body competences in the area of posture and antigravity coordination. Developmental milestones should occur in stages, both in the group of healthy children and in the group of sick children [9,19,29,40]. Concomitant congenital disorders or diseases may change the quality and rate of development [1,20,21,22]. The relation between persistent primitive reflexes and the development of scoliosis and motor coordination disorders has been described in the literature [32]. The timing of developmental milestones, such as age at first walking, is associated with later diagnoses of neurodevelopmental disorders. These findings suggest that early motor milestone achievement is associated with genetic liability for ADHD and autism in the general population. Attention deficit hyperactivity disorder was found to be associated with earlier age at first walking in both males and females [16]. However, there are no data on the mechanisms stimulating this process [31,32]. Data from different checklists do not match and do not provide automatic guidelines for rehabilitation [39]. Therefore, it was necessary to assess which environmental conditions may interfere most with the development of posture and coordination.

In accordance with the original hypothesis, the results confirmed that the quality of postural, coordination, and reflex functions in adolescence shows a high correlation with the course of developmental milestones in the first year of life. In both study groups, the prevalence of postural defects, coordination disorders, and persistent primitive reflexes was found. Statistically significant increases in these occurred in the study group (Figure 2). The intensity of these features at the age of 9–14 correlates with the course of developmental stages in the first year of life. Statistical analysis did not find any significant differences between groups in terms of the number of perinatal diseases and Apgar scores. This confirms what is shown in the data from the literature, i.e., apart from intraventricular hemorrhages, hyperbilirubinemia, and maternal depression, perinatal abnormalities do not determine abnormal development [22]. 

The statistical analysis of development stages as well as the test results from adolescence confirm the above-mentioned hypothesis, namely, that the passive and premature verticalization of the infants shortens the duration and quality of development stages. Children who are seated before they have time to achieve this function independently do not acquire sufficient postural coordination by the time they are assessed at the age of 9–14 years. Accumulated primitive reflexes are visible as well. Statistically, these children often skip the crawling stage and assume positions on straightened lower limbs earlier. They achieve subsequent stages of verticalization in accelerated terms using primitive/tonic and extensor muscle tension. Shorter durations of the stages result in worse effects in the form of immature and distorted milestones or their omission. Compensatory prolongation of development may provide a chance to compensate disorders. The hypothesis of the stimulating effect of gravity on psycho-motor development has been confirmed. The antigravitational work of muscles involving the vestibular organ stimulated by gravity is probably the main factor integrating spinal, subcortical, and cortical neural networks [2]. In both groups, it was observed that the faster the infant’s passive verticalization and sitting were, the more frequent the skipping of milestones and the greater the accumulation of persistent primitive reflexes became. Generally accepted patterns of infant care and the impact of correctable environmental factors on a child’s further development require further research. Early diagnostics and therapy and the replacement of pathological patterns by more regular ones are necessary [1,20,41,42].

Children with developmental delays can achieve subsequent developmental milestones at their own pace, provided that the nervous tissue has not been structurally damaged and the optimal conditions for free motor skills and verticalization are created. A much longer period of maturation of the nervous system is then needed, as is rehabilitation therapy, adequate to the disorder, that does not passively force subsequent stages of upright positioning [16,18,19]. Adopting stable activating positions appropriate to the stage of development and the possibility of free, personalized, or directed motor skills in therapy will stimulate the body to achieve subsequent milestones [5,19,20,27]. The infant cannot maintain balance, and so passive and premature upright positioning, causing tonic postural reflexes, may perpetuate the disharmonious state of coordination. Performing intense physical exercise, including sports, without corrective therapy may retain impaired coordination, primary reflexes and asymmetry in children and adolescents with abnormal muscle tone [32,42]. If cases of central coordination disorders and pervasive developmental disorders result from a large accumulation of primitive reflexes, conducting therapy based on developmental kinesiology may be crucial to improving the integrity of the neuromuscular, cognitive, and social systems of these subjects [2,31,42]. In cases of permanent neurological deficits, assistive devices for upright positioning can be used to facilitate the performance of motor tasks [43,44]. However, it was found that children with cerebral palsy had better functional performances and more precise movements when they achieved greater postural stabilization of the trunk, head, and pelvis [45]. Therefore, in a long-term rehabilitation program, motor skills should be stimulated more in positions that activate the formation of developmental milestones rather than using passive verticalizing devices. The waiting time for subsequent motor and coordination functions in cases of neurological deficits and the determination of the moment when compensatory possibilities have been exhausted may be debatable [46,47,48].

During infancy, it is reasonable for the child to be carefully observed by parents who know the basic stages of the infant’s development and, in case of doubt, obtain the opinion of a medical specialist [14,20,42]. Educating the family regarding care for an infant may be a decisive element of medical supervision [10,11]. The nursing, positioning, and rehabilitation of children in this most important period of life should take place in positions consistent with the current stage of development [2,19,27,42]. The conclusions from this study may contribute to greater integration of the medical community in systematizing views on how to carry and care for infants at individual stages of development. The use of retrospective assessments of the course of milestones described in medical records, supplemented by descriptions from parents, is a limitation of this study. Correlation-based assessment of the remaining features studied was also not performed. Further research in this direction is needed. The study conducted may contribute to reducing the frequency of permanent posture defects, coordination disorders, and related pain dysfunctions in childhood, adolescence, and adulthood [2,49,50,51].

## 5. Conclusions

This study confirmed the wide prevalence of postural defects and coordination disorders among children and adolescents. 

The study’s results showed a close correlation between the development of infant milestones and the quality of body posture and coordination observed at the age of 9–14 years. Independent verticalization on a stable and horizontal surface stimulated the optimal transition of developmental stages and the achievement of subsequent developmental milestones. 

The developmentally incorrect positioning of infants may perpetuate primitive reflexes. Passive and accelerated vertical positioning and sitting in the first year of life and the use of loungers, baby carriers, and walkers may be associated with developmental delay and postural and motor coordination disorders. 

No significant differences were observed between the study groups regarding the number of comorbidities in the perinatal period and Apgar score, which confirms previous reports that the above-mentioned risk factors do not determine abnormal development.

Proper care and the creation of space for the free motor development of infants from birth may be important conditions for proper development in the first year of life. 

It seems advisable to develop and test a screening test, assessing developmental milestones for pediatricians and parents of infants. This would also provide guidance on care and rehabilitation techniques, depending on the stage of development achieved and any comorbidities.

Further studies are needed on the effect of premature verticalization on the development of reflex and coordination functions.

## Figures and Tables

**Figure 1 children-11-01071-f001:**
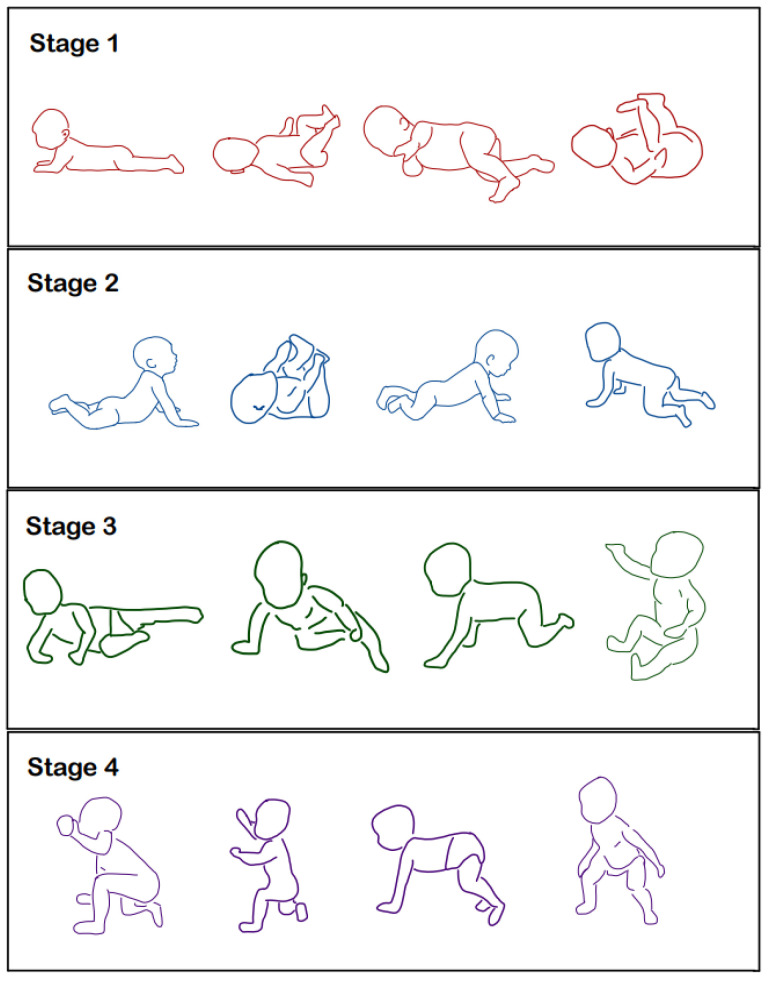
Selected developmental positions in the first year of life according to the principles of developmental kinesiology DNS (dynamic neuromuscular stabilization) [2].

**Figure 2 children-11-01071-f002:**
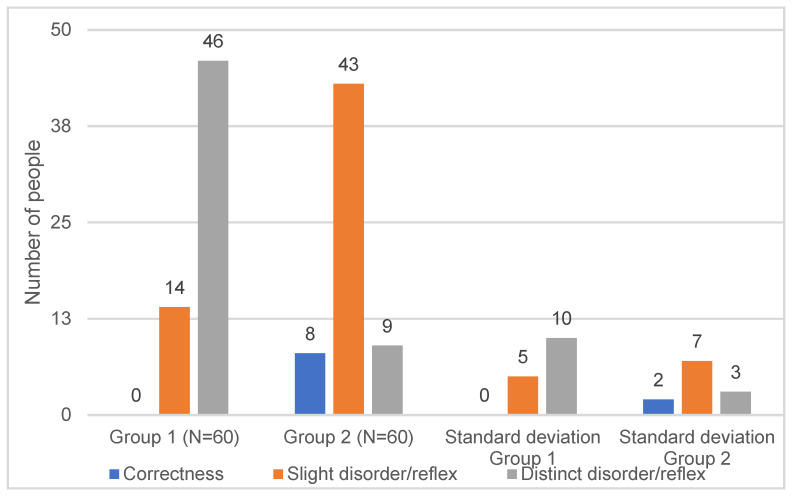
Distribution of body posture assessment results in standing position in group 1 (N = 60) and group 2 (N = 60).

**Table 1 children-11-01071-t001:** Study questionnaire. Abbreviations: ATNR = asymmetric tonic neck reflex; STNR = symmetric tonic neck reflex; DNS = dynamic neuromuscular stabilization (diagnostic and therapeutic rehabilitation method based on developmental kinesiology).

Research Chart	
1. General health status	
2. Assessment of body posture in a standing position	
3. Functional tests acc. to motor development charts of the DNS concept for the following periods:	a. 0–3 months;
	b. 3–6 months;
	c. 6–9 months;
	d. 9–12 months.
4. One-leg stand test	
5. Alternation test	
6. Test assessing the presence of asymmetric tonic neck reflex (ATNR)	
7. Test assessing the presence of Symmetric Tonic Neck Reflex (STNR)	
8. Romberg test with eyes open and closed (with the use of posturography)	
9. Feet assessment using posturography	
10. Medical history and assessment of medical record regarding the following:	a. The presence of risk factors in the perinatal period;
	b. The number of Apgar scores;
	c. The way of carrying and the care and functions, achieved in the period of 0–3 months (body symmetry, joint centralization);
	d. The way of carrying and the care and functions, achieved in the period of 3–6 months (coordinated rotation);
	e. The way of carrying and care and functions, achieved in the period of 6–9 months (all-fours crawling, oblique sit);
	f. The way of carrying and care and functions achieved in the period of 9–12 months (active sitting straight, standing up, walking);
	g. Completed rehabilitation treatment (comprehensive);
	h. The diagnosis of hip dysplasia;
	i. Used rehabilitation/orthopedic aids (e.g., abduction braces);
	j. Comorbidities of developmental age;
	k. Using passive aids (bouncers, seats, and walkers, etc.).

**Table 2 children-11-01071-t002:** Characteristics of the investigated groups.

Parameter	Group 1 (*n* = 60)	Group 2 (*n* = 60)
Age		
Mean value	10.85	10.8
Standard deviation (SD)	1.84	1.59
Minimum value (Min)	8	8
Lower quartile (Q1)	9	10
Median (Q2)	10	11
Upper quartile (Q3)	12	12
Maximum value (Max)	14	14
Gender:		
F	24 (40%)	22 (36.67%)
M	36 (60%)	38 (63.33%)
Shapiro–Wilk test for age normality	normal shape (mesokurtic distribution)	normal shape (mesokurtic distribution)

**Table 3 children-11-01071-t003:** Spearman’s rank correlations calculated for data from the individual stages of development in the first year of life and the results of adequate functional tests performed in adolescence.

**Group 1**	**Assessment of Body Posture in Standing Position**	**Functional Tests According to the Motor Development Charts of the DNS Concept for the Period**				**Assessment Test**		
		**0–3 Months**	**3–6 Months**	**6–9 Months**	**9–12 Months**	**Alternation** **Test**	**ATNR**	**STNR**
Way of carrying and care, functions achieved in the period of 0–3 months (body symmetry, joint centralization)	ns	ns	ns	ns	ns	ns	ns	ns
Way of carrying and care, functions achieved in the period of 3–6 months (coordinated rotation)	0.316	0.209	0.964	0.483	0.314	0.253	ns	ns
Way of carrying and care, functions achieved in the period of 6–9 months (all-fours crawling, oblique sit)	0.333	0.223	0.484	0.871	0.417	0.243	ns	0.208
Way of carrying and care, functions achieved in the period of 9–12 months (active sitting straight, standing up, walking)	0.387	0.234	0.225	0.389	0.806	ns	0.289	0.380
**Group 2**	**Assessment of Body Posture in Standing Position**	**Functional Tests According to The Motor Development Charts Of The DNS Concept For The Period**				**Assessment Test**		
		**0–3 Months**	**3–6 Months**	**6–9 Months**	**9–12 Months**	**Alternation Test**	**ATNR**	**STNR**
Way of carrying and care, functions achieved in the period of 0–3 months (body symmetry, joint centralization)	0.308	0.748	0.977	0.323	0.438	0.431	0.591	0.484
Way of carrying and care, functions achieved in the period of 3–6 months (coordinated rotation)	0.428	0.609	0.862	0.477	0.607	0.473	0.601	0.503
Way of carrying and care, functions achieved in the period of 6–9 months (all-fours crawling, oblique sit)	0.351	0.329	0.334	0.698	0.359	0.385	0.458	0.438
Way of carrying and care, functions achieved in the period 9–12 months of (active sitting straight, standing up, walking)	0.398	0.325	0.505	0.513	0.740	0.499	0.553	0.399

ns: no significant.

**Table 4 children-11-01071-t004:** Correlations between developmental periods of the first year of life and dependent developmental positions at the age of 9–14 years.

**Group 1.**	**Functional Tests According to the Motor Development Charts of the DNS Concept for the Period**			
	**0–3 Months**	**3–6 Months**	**6–9 Months**	**9–12 Months**
Way of carrying and care, functions achieved in the period of 0–3 months (body symmetry, joint centralization)	ns			
Way of carrying and care, functions achieved in the period of 3–6 months (coordinated rotation)		0.964		
Way of carrying and care, functions achieved in the period of 6–9 months (all-fours crawling, oblique sit)			0.871	
Way of carrying and care, functions achieved in the period of 9–12 months (active sitting straight, standing up, walking)				0.806
**Group 2**	**Functional Tests According to the Motor Development Charts of the DNS Concept for the Period**			
	**0–3 Months**	**3–6 Months**	**6–9 Months**	**9–12 Months**
Way of carrying and care, functions achieved in the period of 0–3 months (body symmetry, joint centralization)	0.748			
Way of carrying and care, functions achieved in the period of 3–6 months (coordinated rotation)		0.862		
Way of carrying and care, functions achieved in the period of 6–9 months (all-fours crawling, oblique sit)			0.698	
Way of carrying and care, functions achieved in the period of 9–12 months (active sitting straight, standing up, walking)				0.740

**Table 5 children-11-01071-t005:** Statistical differences between the investigated groups.

Test		
1. General health status		t = 3.03; *p* = 0.002 d = 0.56
2. Assessment of body posture in a standing position		t = 8.55; *p* < 0.001 d = 1.57
3. Functional tests according to motor development charts of the DNS concept for the following periods:	a. 0–3 months;	t = 6.95; *p* < 0.001 d = 1.28
	b. 3–6 months;	t = 7.46; *p* < 0.001 d = 1.37
	c. 6–9 months;	t = 6.00; *p* < 0.001 d = 1.10
	d. 9–12 months.	t = 4.19; *p* < 0.001 d = 0.77
4. One-leg stand test		t = 6.56; *p* < 0.001 d = 1.21
5. Alternation test		t = 4.02; *p* < 0.001 d = 0.74
6. Test assessing the presence of asymmetric tonic neck reflex (ATNR)		t = 3.66; *p* < 0.001 d = 0.67
7. Test assessing the presence of symmetric tonic neck reflex (STNR)		t = 4.27; *p* < 0.001 d = 0.78
8. Romberg test with eyes open and closed (with the use of posturography)		t = 5.40; *p* < 0.001 d = 0.99
9. Feet assessment using posturography		t = 4.55; *p* < 0.001 d = 0.84
10. Medical history and assessment of medical record regarding the following:	a. The presence of risk factors in the perinatal period;	n. s.
	b. The number of Apgar scores;	n. s.
	c. The way of carrying and care and functions achieved in the period 0–3 months (body symmetry, joint centralization);	t = 6.63; *p* < 0.001 d = 1.22
	d. The way of carrying and care and functions achieved in the period 3–6 months (coordinated rotation);	t = 6.72; *p* < 0.001 d = 1.23
	e. The way of carrying and care and functions achieved in the period 6–9 months (all-fours crawling, oblique sit);	t = 5.52; *p* < 0.001 d = 1.01
	f. The way of carrying and care and functions achieved in the period 9–12 months (active sitting straight, standing up, walking);	t = 4.61; *p* < 0.001 d = 0.85
	g. Completed rehabilitation treatment (comprehensive);	n. s.
	h. The diagnosis of hip dysplasia;	t = 2.95; *p* = 0.002 d = 0.54
	i. Used rehabilitation/orthopedic aids (e.g., abduction braces);	n. s.
	j. Comorbidities of developmental age;	n. s.
	k. Using passive aids (bouncers, walkers, etc.).	t = 1.96; *p* = 0.026 d = 0.54

n.s.: no significant.

**Table 6 children-11-01071-t006:** Characteristics of the age of assuming a sitting position and walking.

Parameter	Group 1 (*n* = 60)	Group 2 (*n* = 60)
Child’s sitting age:		
Mean value	5.92 month	6.20 month
Standard deviation (SD)	1.65 month	1.60 month
Minimum value (Min)	3 month	4 month
Lower quartile (Q1)	5 month	5.75 month
Median (Q2)	6 month	6 month
Upper quartile (Q3)	6 month	7 month
Maximum value (Max)	12 month	10 month
Walking age:		
Mean value	11.52 month	11.87 month
Standard deviation (SD)	1.42 month	1.37 month
Minimum value (Min)	7 month	9 month
Lower quartile (Q1)	11 month	11 month
Median (Q2)	11.5 month	12 month
Upper quartile (Q3)	12 month	12 month
Maximum value (Max)	16 month	18 month

## Data Availability

The original contributions presented in the study are included in the article, further inquiries can be directed to the corresponding author/s.

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
