# Peer review of "The Assessment of Postural–Motor, Coordination, and Reflex Functions in Children and Adolescents with a History of Premature Verticalization and Ontogeny Disorders in Their First Year of Life"

_children, 2024, doi:10.3390/children11091071_

Round 1
Reviewer 1 Report
Comments and Suggestions for Authors
Dear Author;
A different perspective is presented in the study, but some points are not understood.
Abstract
120 children were divided into two groups and the second group consisted of volunteers. Both groups should volunteer to participate in the study.
How was the first-year data of 120 children aged 9-14 obtained?
Introduction
What is the hypothesis of the study?
Methods
What were the demographic characteristics of the children?
"The first group included patients with diagnosed faulty posture under the care of a posture treatment outpatient clinic." What was the posture disorder? Posture disorder, kyphosis, scoliosis...etc?
"A group of 120 subjects, aged 9-14 years, was examined, by performing functional tests and retrospective analysis of their development until age 1. The patients were divided into 2 groups of 60 each." Instead, 60 children under treatment for postural disorders and 60 children of similar age without postural disorders were evaluated. A statement like this would be more descriptive. From your writing, it is understood that there were 120 children at the beginning and they were divided into two groups. However, you have created a control group for children with postural disorders at the same age.
Asking the family about the development of children aged 9-14 years in the first year would be a subjective evaluation. The family may have difficulty in remembering the past time. How did you find a solution for this?
Children without neurological problems were included in the study. Why were STNR and ATNR evaluated in these children?
Results
How the "research chart" questions were scored.
The p-value should be added to the tables.
The expression "mths" in the tables should be changed to "months".
"Month" should be added as a unit at the end of "Child's sitting and walking age".
"lack of a control group of adolescents with normal postural and coordination functions are limitations of the study." Isn't the control group in the method a child without postural disorder?
Discussion
What is the scientific significance of the study results?
References
References must be checked according to the author’s guidelines.
Author Response
Dear Reviewer,
thank you very much for your insightful comments.
Abstract
120 children were divided into two groups and the second group consisted of volunteers. Both groups should volunteer to participate in the study.
Both groups of patients under study volunteered. The first study group consisted of people under the care of the Posture Treatment Clinic, while the control group consisted of healthy people who were not treated at the Posture Treatment Clinic and volunteered for the study.
How was the first-year data of 120 children aged 9-14 obtained?
Retrospective data on the development of milestones and care techniques during the first year of life were obtained from medical records and interviews with caregivers during this period. Visual charts of child development during the first year of life were used to assist caregivers in answering questions. Photographs taken during this time period were also analyzed to assess the achievement of motor milestones.
Introduction
What is the hypothesis of the study?
The hypothesis of the study is that children achieve subsequent developmental milestones through independent verticalization and free motor skills. By providing the child with stable supporting positions appropriate to the developmental age achieved and a safe space, we enable the development of support and extension functions. The supposed mechanism integrating immature spinal, subcortical and cortical neural networks is active verticalization with the leading role of the vestibular organ. Passive positioning and carrying of an infant in a position that is too vertical in relation to the stage of development may stimulate primitive reflex reactions and perpetuate them due to the developmental immaturity of the balance system. The study design was also based on the hypothesis that subsequent stages of a child's development in the first year of life shape the basic structures and functions of the body for life, which will then only be expanded and improved. For this reason, young people from a wide age range of 9-14 years were qualified for the study to include the period of accelerated and slowed body growth. The study was to confirm the extent to which postural functions, coordination and reflexes are a reflection of developmental stages in the first year. This thesis was confirmed, therefore, errors in infant care in the first year of life can leave a developmental trace in the quality of posture and sensorimotor coordination for the entire individual's life. There are many adolescent and adult diseases of idiopathic etiology, in which improper antigravity management is visible. The developmental nature of these disorders can contribute to finding causal therapeutic methods.
Methods
What were the demographic characteristics of the children?
The study participants in group 1 were recruited from patients aged 9-14 who were referred to a posture treatment clinic due to body shape and movement coordination disorders, including scoliosis, severe kyphosis, valgus or varus deformity of the limbs. Volunteers in group 2 were people of the same age, who were supposed to be healthy, practicing sports, recruited from local primary schools.
"The first group included patients with diagnosed faulty posture under the care of a posture treatment outpatient clinic." What was the posture disorder? Posture disorder, kyphosis, scoliosis...etc?
Participants in group 1 of the study were recruited from among patients who were referred to a posture treatment clinic due to body shape disorders and motor coordination disorders, including scoliosis, severe kyphosis, valgus or varus deformity of the limbs.
"A group of 120 subjects, aged 9-14 years, was examined, by performing functional tests and retrospective analysis of their development until age 1. The patients were divided into 2 groups of 60 each." Instead, 60 children under treatment for postural disorders and 60 children of similar age without postural disorders were evaluated. A statement like this would be more descriptive. From your writing, it is understood that there were 120 children at the beginning and they were divided into two groups. However, you have created a control group for children with postural disorders at the same age.
The study assessed 60 children aged 9-14 treated in a posture defect clinic and 60 healthy children in the same age range as a control group, in whom no significant posture disorders were previously diagnosed. A total of 120 people aged 9-14 were examined, performing functional tests and a retrospective analysis of their development until the end of the first year of life.
Asking the family about the development of children aged 9-14 years in the first year would be a subjective evaluation. The family may have difficulty in remembering the past time. How did you find a solution for this?
Retrospective analysis of development in the first year of life of a person aged 9-14 is partly subjective, as described in the article. The solution to the problem was to ask detailed questions to parents, evaluate photographs taken during this period, and analyze records in medical documentation regarding developmental milestones achieved. The use of visual child development tables according to the DNS Kolar concept facilitated both the assessment of development and the performance of functional tests. People adopted after the age of 1 were not qualified for the study. In most cases, parents were supported by photographs taken during this period, which is not currently a problem due to the development of digital photography in the world.
Children without neurological problems were included in the study. Why were STNR and ATNR evaluated in these children?
STNR and ATNR patterns are observed and assessed in the first year of life as physiological reflex reactions of the developing child's organism, which should be integrated in the first year of life. Therefore, these reflex patterns should not be visible at the age of 9-14 years. Persisting until this period may cause disorders of central coordination of movement and body posture. The study showed that many neurologically healthy people present partially persistent primitive reflexes as functional disorders. The publication cited works describing the characteristics of primitive reflexes and the consequences of their lack of full integration.
Results
How the "research chart" questions were scored.
The results of the tests and procedures were entered into the research card using the following scoring:
1. regularity;
2. slight disturbance/reflex;
3. marked disturbance/reflex.
In the case of a score of 2 or 3, an additional description of the abnormalities was made with the active participation of the examined person and his/her parent. The scoring assessing the quality of development in the individual stages of the first year of life was based on the assessment of information obtained from the parents and familiarization with the patients' medical records
The p-value should be added to the tables.
The level of statistical significance p (p-value) is included in Table 5.
The expression "mths" in the tables should be changed to "months".
The word "months" has been corrected as recommended.
"Month" should be added as a unit at the end of "Child's sitting and walking age".
The word "month" was entered in Table 6 as recommended.
"lack of a control group of adolescents with normal postural and coordination functions are limitations of the study." Isn't the control group in the method a child without postural disorder?
The control group included people who had not been previously diagnosed with postural defects. During the study, some of these people showed postural defects and mild coordination disorders in a statistically lower degree of severity. This study did not design an assessment of a uniform control group of people who only present optimal body structures and functions in order to track the quality of development of these people in the first year of life. The frequency of postural disorders indicates that it would be difficult to recruit such a number of people with ideal coordination. However, conducting such a study was a good challenge.
Discussion
What is the scientific significance of the study results?
The results of the study challenge the current views on the necessity of passive and frequent verticalization and seating of infants, especially in the first 6 months of life. Although this habit helps to calm the child temporarily, it may reinforce negative reflex patterns in further development. Early use of armchairs, carriers and walkers helps in caring for the infant, but limits the free development of coordination and may burden and deform growth cartilages. The study emphasizes the close connection between the course of developmental milestones in the first year of life and health, fitness and body posture formation in later periods of life. Perinatal problems have an impact, but do not determine incorrect further development. Statistical analysis of the study confirms that passive and premature verticalization of infants reinforces primitive reflexes and may have a negative impact on the psychomotor development of children. The study may contribute to greater integration of the medical community in systematizing views on the method of carrying and care at individual stages of development. These conclusions may contribute to reducing the incidence of persistent postural defects, coordination disorders and inextricably linked pain dysfunctions and deformations of the musculoskeletal system. The study may also influence the selection of rehabilitation techniques appropriate to the child's developmental age. Similarly, rehabilitation of adolescent and adult diseases may take into account the effectiveness of developmental positions activating central coordination. A developmental approach to effective rehabilitation treatment requires taking into account the duration of training, as well as each stage of development ending with the achievement of another milestone. The study may also be one of the voices explaining the cause of different developmental paths in children.
References
References must be checked according to the author’s guidelines.
The quality and order of references were checked
Reviewer 2 Report
Comments and Suggestions for Authors
The present study is original and relevant to the field of this journal, however there are points that need revisions, such as language, methodological and structural issues. Structural and methodological issues are the main focus : RESEARCH/statistical HYPOTHESES are recommended to be included and stated clearly and according to examined variables. In this mode the introduction, results, discussion and conclusions sections are recommended to be organized accordingly. There are also comments inside the text.
General comments:
Abstract:
The abstract is more clearly organized that the other sections of the manuscript. However, it needs revisions according to manuscript.
Introduction:
1. Although the literature review is brief and well-written, it is recommended that the authors should clarify better the rationale of the study in order to become more robust (lines 64-68… There are inconsistencies … it is not enough to explain and emphasize the importance of your study. In addition you use only one reference).
2. RESEARCH/statistical HYPOTHESES should be stated clearly before the purpose according to variables examined and the introduction section should be structured accordingly.
3. It is recommended that the authors present a better review of previous studies regarding the measurements they used (lines 75-80)
Materials and Methods:
1. A better description of the participants is needed. They were patients as the authors mentioned, it is advisable both groups to be described in detail (developmental delays, disorders, disabilities, cognitive development…if groups participated in physical activity or sports etc.). A table depicting both groups’ exact characteristics would be more illustrative. In addition, the age range (9-14 years old) is too extended. Did the authors calculate the developmental age of both groups? How did they manage any inconsistencies in order to produce valid results? Please explain in detail the reason for using this age range and the role of developmental age. Maybe a group of 9-11and 11 months and a group of 12 to 14 and 11 months would be more concise. Please give your rationale.
2. It is advisable that validity and reliability issues regarding the tests / questionnaires should be addressed.
3. Regarding the use of “Romberg” test for balance evaluation the authors should give a rationale taking in consideration not only the chronological and developmental age but the disorder/disability of the participants. A review of this measurement coupled with the specific characteristics of both groups should be included in the introduction section, too.
4. The procedure is not included in the methods section. It is recommended the 1st paragraph of the method and materials section to be placed as a 1st paragraph of the procedure section following the description of the tests used. It should also be added a paragraph regarding the procedure in detail.
Statistical analyses/Results:
The results section is confused and tables and figures need revision. In addition the statistical analyses and results section should be organized according to research hypotheses.
Table 5: is the table depicting a t-test? This analysis should be mentioned in the statistical analysis section.
Discussion:
Discussion should be started by stating the purpose of the study. It is advisable that the authors should structure the main text of discussion according to RESEARCH/statistical HYPOTHESES and findings. It is recommended to be revised thouroughly and precisely e.g. Accodring to the first hypothesis the results showed/revealed… discuss each result according to each hypothesis… with regard to literature. Recommendations for future research should be added.
If the authors wish to publish this manusctript in this journal should take into account the above general recommendations for revisions and also comments inside the text, because their study has the potential to be scientifically valid, technically sound and make an original contribution to the literature.

Comments on the Quality of English LanguageThe use of past tenses is recommended for studies conducted in the past (e.g. instead of are …were when results are described in abstract and in the main text and other where it is applicable). It is advisable 3rd person in manuscript and passive voice to be used (comments are inside the text).
Author Response
Dear Reviewer,
thank you very much for your insightful comments.
Abstract:
The abstract is more clearly organized that the other sections of the manuscript. However, it needs revisions according to manuscript.
The abstract has been revised to make the study topic and results clearer.
Introduction:
1. Although the literature review is brief and well-written, it is recommended that the authors should clarify better the rationale of the study in order to become more robust (lines 64-68… There are inconsistencies … it is not enough to explain and emphasize the importance of your study. In addition you use only one reference).
2. RESEARCH/statistical HYPOTHESES should be stated clearly before the purpose according to variables examined and the introduction section should be structured accordingly.
3. It is recommended that the authors present a better review of previous studies regarding the measurements they used (lines 75-80)
In the introduction section, a long paragraph was added explaining the developmental background of the issue under study, the course of developmental milestones, and the reason for designing this study. In accordance with the recommendation, the reason and role of balance assessment were described, including the Romberg Test for the development of the postural and reflex functions under study. Several references from the literature describing the current knowledge about the relationship between balance development and the integration of primitive reflexes and the development of postural coordination were cited. The study may shed new light on the cause of different developmental trajectories of children considered healthy and the developmental possibilities of sick children. As recommended, the paragraph concerning the description of the reflex functions examined was moved.
The hypothesis of the study is that children achieve subsequent developmental milestones through independent verticalization and free motor skills. By providing the child with stable supporting positions appropriate to the developmental age achieved and a safe space, we enable the development of support and extension functions. The supposed mechanism integrating immature spinal, subcortical and cortical neural networks is active verticalization with the leading role of the vestibular organ. Passive positioning and carrying of an infant in a position that is too vertical in relation to the stage of development may stimulate primitive reflex reactions and perpetuate them due to the developmental immaturity of the balance system. The study design was also based on the hypothesis that subsequent stages of a child's development in the first year of life shape the basic structures and functions of the body for life, which will then only be expanded and improved. For this reason, young people from a wide age range of 9-14 years were qualified for the study to include the period of accelerated and slowed body growth. The study was to confirm the extent to which postural functions, coordination and reflexes are a reflection of developmental stages in the first year. This thesis was confirmed, therefore, errors in infant care in the first year of life can leave a developmental trace in the quality of posture and sensorimotor coordination for the entire individual's life. There are many adolescent and adult diseases of idiopathic etiology, in which improper antigravity management is visible. The developmental nature of these disorders can contribute to finding causal therapeutic methods.
Materials and Methods:
1. A better description of the participants is needed. They were patients as the authors mentioned, it is advisable both groups to be described in detail (developmental delays, disorders, disabilities, cognitive development…if groups participated in physical activity or sports etc.). A table depicting both groups’ exact characteristics would be more illustrative. In addition, the age range (9-14 years old) is too extended. Did the authors calculate the developmental age of both groups? How did they manage any inconsistencies in order to produce valid results? Please explain in detail the reason for using this age range and the role of developmental age. Maybe a group of 9-11and 11 months and a group of 12 to 14 and 11 months would be more concise. Please give your rationale.
The study assessed 60 children aged 9-14 treated in a posture defect clinic and 60 healthy children in the same age range as a control group, in whom no significant posture disorders were previously diagnosed. A total of 120 people aged 9-14 were examined, performing functional tests and a retrospective analysis of their development until the end of the first year of life.
The study participants in group 1 were recruited from patients aged 9-14 who were referred to a posture treatment clinic due to body shape and movement coordination disorders, including scoliosis, severe kyphosis, valgus or varus deformity of the limbs. Volunteers in group 2 were people of the same age, who were supposed to be healthy, practicing sports, recruited from local primary schools.
The control group included people who had not been previously diagnosed with postural defects. During the study, some of these people showed postural defects and mild coordination disorders in a statistically lower degree of severity. This study did not design an assessment of a uniform control group of people who only present optimal body structures and functions in order to track the quality of development of these people in the first year of life. The frequency of postural disorders indicates that it would be difficult to recruit such a number of people with ideal coordination. However, conducting such a study was a good challenge.
The study design was also based on the hypothesis that the subsequent stages of a child's development in the first year of life shape the basic structures and functions of the body for life, which will then only be expanded and improved. For this reason, young people from a wide age range of 9-14 years were qualified for the study, to include a period of accelerated and decelerated body growth. The study was to confirm the extent to which postural functions, coordination and reflexes are a reflection of the stages of development in the first year.
2. It is advisable that validity and reliability issues regarding the tests / questionnaires should be addressed.
The study design assumed subjective evaluation of the tests. However, in order to improve the validity and reliability of the tests, the study was based on detailed performance of functional tests, conducted by the examining physician in cooperation with the participants and their caregivers. Scores of the performed tests from 1 to 3 were given jointly after discussion. Visual charts of motor development were used to objectify the quality of the performed developmental positions and postural tests and to remind the time of achieving developmental milestones (Fig. 1). In addition, the retrospective assessment of the child's developmental milestones used data from submitted medical records, archival photographs and information collected from the child's caregivers. To objectify the results of the balance assessment in the Romberg test and the quality of foot loading, the Free-Step digital posturograph was used.
3. Regarding the use of “Romberg” test for balance evaluation the authors should give a rationale taking in consideration not only the chronological and developmental age but the disorder/disability of the participants. A review of this measurement coupled with the specific characteristics of both groups should be included in the introduction section, too.
In the point concerning the introductory section, in accordance with the recommendation, the reason and role of balance assessment were described, including, among others, the Romberg Test for the development of the postural and reflex functions under examination. Several references from the literature describing the current knowledge about the relationship between balance development and the integration of primitive reflexes and the development of postural coordination were cited.
Assessment of body balance by performing, for example, the one-leg standing test, the alternation test and the Romberg test using a posturograph allows for the assessment of the maturity of the nervous system [4,18,19]. Motor development is characterized by the disappearance of primitive reflexes and the acquisition of balance competence. There is a proven relationship between the quality of balance, the integration of primitive reflexes and psychomotor and educational abilities at school age [31,32]. Proper stimulation of the vestibular organ affects the development of vision and body posture [5,36].
4. The procedure is not included in the methods section. It is recommended the 1st paragraph of the method and materials section to be placed as a 1st paragraph of the procedure section following the description of the tests used. It should also be added a paragraph regarding the procedure in detail.
As recommended, the paragraph was moved to the procedures section and the procedures were described in more detail.
Statistical analyses/Results:
The results section is confused and tables and figures need revision. In addition the statistical analyses and results section should be organized according to research hypotheses.
Table 5: is the table depicting a t-test? This analysis should be mentioned in the statistical analysis section.
The results section has been improved to be more readable. Language issues in tables and figures have been corrected. In the statistical analysis section, the t-test has been added.
Discussion:
Discussion should be started by stating the purpose of the study. It is advisable that the authors should structure the main text of discussion according to RESEARCH/statistical HYPOTHESES and findings. It is recommended to be revised thouroughly and precisely e.g. Accodring to the first hypothesis the results showed/revealed… discuss each result according to each hypothesis… with regard to literature. Recommendations for future research should be added.
In accordance with the recommendation, the Discussion section was expanded and reorganized in the following order: description of the study purpose, assumption of the study, and hypotheses that were confirmed. The obtained results were discussed in detail in relation to the adopted hypotheses. Then, conclusions and their reference to child care and developmental rehabilitation practice were presented. The conclusions indicate the need for further research on the impact of the quality of verticalization on health and fitness.
If the authors wish to publish this manusctript in this journal should take into account the above general recommendations for revisions and also comments inside the text, because their study has the potential to be scientifically valid, technically sound and make an original contribution to the literature.
Comments on the Quality of English Language
The use of past tenses is recommended for studies conducted in the past (e.g. instead of are …were when results are described in abstract and in the main text and other where it is applicable). It is advisable 3rd person in manuscript and passive voice to be used (comments are inside the text).
During proofreading, comments contained in the text were taken into account.
Round 2
Reviewer 1 Report
Comments and Suggestions for Authors
Dear Author;
Thank you for making the corrections meticulously. I would like to highlight a few minor points that caught my attention.
1. The percentage distribution of children with postural disorders according to the disorder can be added as demographic information in the results
2. On page 14, a paragraph is bolded.
3. The conclusion can be written as a paragraph, not as a bullet point.
4. In reference 1, the full article title is capitalized. I think it is caused by a program like "endnote".
I wish you to continue your work.
Author Response
Dear Author;
Thank you for making the corrections meticulously. I would like to highlight a few minor points that caught my attention.
Dear Reviewer,
thank you very much for all your valuable comments. The manuscript has been improved according to your suggestions.
1. The percentage distribution of children with postural disorders according to the disorder can be added as demographic information in the results
Thank you for the suggestion. It was moved.
2. On page 14, a paragraph is bolded.
Thank you for the suggestion. It was corrected.
3. The conclusion can be written as a paragraph, not as a bullet point.
Thank you for the suggestion. It was corrected.
4. In reference 1, the full article title is capitalized. I think it is caused by a program like "endnote".
Thank you for the suggestion. It was corrected.
I wish you to continue your work.